# Reactogenicity to the mRNA-1273 Booster According to Previous mRNA COVID-19 Vaccination

**DOI:** 10.3390/vaccines10081217

**Published:** 2022-07-29

**Authors:** Oleguer Parés-Badell, Ricardo Zules-Oña, Lluís Armadans, Laia Pinós, Blanca Borrás-Bermejo, Susana Otero, José Ángel Rodrigo-Pendás, Martí Vivet-Escalé, Yolima Cossio-Gil, Antònia Agustí, Cristina Aguilera, Magda Campins, Xavier Martínez-Gómez

**Affiliations:** 1Servei de Medicina Preventiva i Epidemiologia, Vall d’Hebron Hospital Universitari, Vall d’Hebron Barcelona Hospital Campus, Passeig Vall d’Hebron 119-129, 08035 Barcelona, Spain; rgzules.girona.ics@gencat.cat (R.Z.-O.); lluis.armadans@vallhebron.cat (L.A.); laia.pinos@vallhebron.cat (L.P.); blanca.borras@vallhebron.cat (B.B.-B.); susana.otero@vallhebron.cat (S.O.); josean.rodrigo@vallhebron.cat (J.Á.R.-P.); marti.vivet@vallhebron.cat (M.V.-E.); xavier.martinez@vallhebron.cat (X.M.-G.); 2Grup de recerca de Epidemiologia i Salut Pública, Vall d’Hebron Institut de Recerca (VHIR), Vall d’Hebron Hospital Universitari, Vall d’Hebron Barcelona Hospital Campus, Passeig Vall d’Hebron 119-129, 08035 Barcelona, Spain; mcampins@vhebron.net; 3Unitat Docent Vall d’Hebron, Universitat Autònoma de Barcelona, 08193 Bellaterra, Spain; 4Department Information Systems and Decision Support, Hospital Universitari Vall d’Hebron, 08035 Barcelona, Spain; yolima.cossio@vallhebron.cat; 5Research Group of Healthcare System Management, Vall d’Hebron Institut de Recerca (VHIR), 08035 Barcelona, Spain; 6Servei de Farmacologia Clínica, Vall d’Hebron Institut de Recerca (VHIR), 08035 Barcelona, Spain; antonia.agusti@vallhebron.cat (A.A.); cristina.aguilera@vallhebron.cat (C.A.); 7Departament de Farmacologia, Terapèutica i Toxicologia, Universitat Autònoma de Barcelona, 08193 Bellaterra, Spain

**Keywords:** SARS-CoV-2, COVID-19-vaccination, booster dose, mRNA vaccines, adverse reactions, health care workers

## Abstract

The objective of this study was to assess the local and systemic adverse reactions after the administration of a COVID-19 mRNA-1273 booster between December 2021 and February 2022 by comparing the type of mRNA vaccine used as primary series (mRNA-1273 or BNT162b2) and homologous versus heterologous booster in health care workers (HCW). A cross-sectional study was performed in HCW at a tertiary hospital in Barcelona, Spain. A total of 17% of booster recipients responded to the questionnaire. The frequency of reactogenicity after the mRNA-1273 booster (88.5%) was similar to the mRNA-1273 primary doses (85.8%), and higher than the BNT162b2 primary doses (71.1%). The reactogenicity was similar after receiving a heterologous booster compared to a homologous booster (88.0% vs. 90.2%, *p* = 0.3), and no statistically significant differences were identified in any local or systemic reactions. A higher frequency of medical leave was identified in the homologous booster dose group vs. the heterologous booster dose group (AOR 1.45; 95% CI: 1.00–2.07; *p* = 0.045). Our findings could be helpful in improving vaccine confidence toward heterologous combinations in the general population and in health care workers.

## 1. Introduction

The introduction of COVID-19 vaccines has changed the course of the infection worldwide. Several studies have confirmed the efficacy of the COVID-19 vaccine in preventing disease severity including hospitalization and mortality rate [1,2,3]. However, studies have shown that the efficacy of COVID-19 vaccines decreases over time in the immunocompetent population [4,5,6,7].

The emergence of the Delta and Omicron SARS-CoV-2 variants has been associated with an increase in transmissibility and a decrease in naive or vaccine-induced immunity [8,9]. Starting December 2021, there has been an escalation in the amount of cases in Europe including Spain, primarily among the unvaccinated, but also among the vaccinated [10].

In November 2021, the Spanish Public Health authority approved the administration of an mRNA first booster for health care workers [11]. The COVID-19 booster has been proven to be effective and safe; however, some studies have reported that a booster could be more reactogenic [12,13].

In a previous study, we compared the frequency of adverse reactions by vaccine type and history and seriousness of a previous COVID-19 infection after the primary vaccination series [14]. In the present study, the objective was to assess the local and systemic adverse reactions in health care workers (HCW) after the administration of a first booster of mRNA-1273 COVID-19 vaccine according to the type of mRNA vaccine used as a primary series, and a homologous versus heterologous booster.

## 2. Materials and Methods

The administration of a COVID-19 vaccine first booster was offered to all vaccinated health care workers in a tertiary hospital in Barcelona (Hospital Universitari Vall d’Hebron). The hospital serves a population of 1.2 million people and has around 7695 health professionals. The booster campaign started on December 2021 using the mRNA-1273 vaccine. The mRNA-1273 vaccine was used according to the vaccination protocols and was prioritized over the BNT162b2 vaccine because of availability. For the purpose of this study, the booster was defined as a subsequent dose of vaccine administered after a COVID-19 primary series vaccination. Some participants were administered one dose of the mRNA-1273 or BNT162b2 vaccine as a primary series because they had a history of COVID-19 infection [11].

This was a cross-sectional study using an online ad hoc survey that we had already used in a previous study [14]. All health care workers who were vaccinated with the booster were invited to answer a self-reported questionnaire at least 5 days after receiving the dose. The questionnaire was available through corporative mailing and the institutional webpage.

The questionnaire (Appendix A) collected information on age, gender, worker category, history of severe allergic reaction, history of chronic illness, history and seriousness of COVID-19 infection, COVID-19 vaccine type, dates of vaccination, adverse reactions to the first or second dose, adverse reactions to the booster, onset and end of the adverse reactions, need for medical attention, need for medical leave, and potential life-threatening reactions. The history of COVID-19 infection did not take into account whether the infection occurred before or after the primary series vaccination. A history of severe allergic reaction was defined as having suffered a anaphylactic shock or glottis edema [15]. We adapted the Center for Disease Control and Prevention definition of chronic illnesses [16] to include cardiac insufficiency, ischemic heart disease, asthma, diabetes, chronic bronchitis, neurological disease, kidney failure, or chronic liver disease. Voluntarily, the participants could reveal their health record ID code, which allowed us to review the self-reported severe reactions in the participants’ clinical histories.

We calculated the frequency and 95% confidence intervals (95% CI) of adverse reactions to COVID-19 vaccination. The frequency of adverse reactions was compared between the recipients of the BNT162b2 and the mRNA-1273 vaccines, the first and second dose, and the history of a previous COVID-19 infection.

We compared the frequency of adverse reactions following the booster versus the previous vaccine doses. Moreover, we compared the frequency of adverse reactions in those patients who received a homologous booster (previous doses of mRNA-1273 and a booster of mRNA-1273) versus those patients who received a heterologous booster (previous doses of BNT162b2 and a booster of mRNA-1273). For the comparisons, we used Pearson’s chi-squared test and Fisher’s exact test for the categorical variables and the Wilcoxon or Kruskal–Wallis rank sum tests for the continuous variables. We performed a multivariate regression model to calculate the adjusted odds ratio and 95% confidence intervals (95% CI) of medical leave according to the sociodemographic characteristics and medical history variables. The model was adjusted by gender, age, workers’ category, history of allergies, history of chronic illness, history of COVID-19 infection, and COVID-19 vaccine booster.

Data were analyzed using the statistical computing program R. The study was approved by the Vall d’Hebron Ethics Committee (approval code: PR(AG)112/2021).

## 3. Results

A total of 7152 health care workers at our center received a COVID-19 vaccine booster between December 2021 and February 2022. Among those, 1222 (17.0%) responded to the self-reported questionnaire (Table 1). A total of 99.2% of the participants (95% CI: 98.7–99.7%) had received a booster with the mRNA-1273 vaccine. Most participants had received a first vaccine dose with either BNT162b2 (76.4%; 95% CI: 74.1–78.8%) or mRNA1273 (20.1%; 95% CI: 17.9–22.4%); 95% CI: 15.8–20.2%). Our sample included a majority of female participants (81.9%; 95% CI: 79.8–84.1%) and registered nurses (45.3%; 95% CI: 42.5–48.0%).

At least one adverse reaction to the mRNA-1273 booster was reported by 88.5% (95% CI: 86.7–90.3%) of the participants, compared to 85.8% (95% CI 81.4–90.1%) for the mRNA1723 primary doses and 71.1% (95% CI: 68.2–74.0%) for BNT162b2 primary doses (Table 2 and Figure 1). According to the type of adverse reactions, the frequency of each of them after the mRNA-1273 booster was similar compared to the mRNA-1273 primary doses, and higher than the BNT162b2 primary doses. However, fewer participants reported fever after an mRNA-1273 booster (40.3%; 95% CI: 37.5–43.1%) compared to after the mRNA-1273 primary doses (50.0%; 95% CI: 43.8–56.2%). In contrast, more participants reported nausea or vomiting and adenopathy after the mRNA-1273 booster compared to the primary series. Four participants reported a potential life-threatening reaction after the mRNA-1273 booster; nevertheless, after reviewing their medical histories, none of those reports were correct, and one patient, after receiving the booster, required medical assistance for malaise and low-grade fever that only required 24 h observation without hospitalization.

Participants who were vaccinated with a heterologous booster (primary doses of BNT162b2 and a booster of mRNA-1273) did not report a higher frequency of adverse reactions compared to those vaccinated with a homologous booster (primary doses and a booster of mRNA-1273) (Table 3). A total of 90.2% (95% CI: 86.5–94.0%) of the homologous group reported at least one adverse reaction compared to 88.0% (95% CI: 85.9–90.1%) of the heterologous group (*p* = 0.300). The duration of the adverse reactions was the same in both groups (*p* = 0.900). A higher frequency of participants who received a homologous booster dose needed medical leave compared to those who received a heterologous booster dose (25.7%; 95% CI: 19.9–31.4% versus 18.4%; 95% CI: 15.7–21.0%; *p* = 0.016). However, we did not identify any local or systemic adverse reaction that was statistically more frequent in the heterologous or the homologous group. Among the recipients of the BNT162b2 primary doses, 61.7% (95% CI: 58.4–65.0%) reported that the mRNA-1273 booster was more reactogenic. In contrast, among the recipients of the mRNA-1273 primary doses, 46.8% (95% CI: 40.3–53.4%) reported that the mRNA-1273 booster was more reactogenic (*p* < 0.001).

As shown in Table 4, participants with a history of chronic illness had a higher probability of needing medical leave compared to participants without chronic illness (AOR 1.63; 95% CI: 1.03–2.53; *p* = 0.032). Moreover, participants who had received an homologous booster had 45% more probability of needing medical leave compared to participants who had received an heterologous booster (AOR 1.45; 95% CI: 1.00–2.07; *p* = 0.045).

## 4. Discussion

### 4.1. Statement of Principal Findings

Our cross-sectional study on health care workers showed that using a heterologous booster did not increase the local nor systemic reactogenicity compared to using a homologous booster. However, participants who had received a homologous booster reported a higher frequency of medical leave compared to the heterologous booster. No severe adverse reactions or potential life-threatening reactions were reported after booster administration. Moreover, according to our results, vaccination with a mRNA-1273 booster had a similar frequency of most of the local and systemic adverse reactions compared to the primary doses with the same vaccine type. The frequency of adenopathy was higher after the booster compared to the mRNA-1273 primary series.

### 4.2. Strengths and Weaknesses of the Study

Comparing our results with our previous study, we had participants with homogeneous characteristics and both investigations had a similar selection criteria and methodology for obtaining the data collection using an ad hoc questionnaire. The samples of respondents of both studies were similar in terms of gender, age, and workers’ category. Our previous study identified that the mRNA-1273 vaccine group reported more prevalent adverse reactions than the BNT162b2 vaccine including medical leave [14]. These results were similar to the findings in the present study, with an increased frequency of adverse reactions after the mRNA-1273 primary series compared to the BNT162b2 primary series. The fact that we obtained coherent results in both studies may indicate an adequate internal and external validity. Contrary to our results regarding the primary series, where a history of previous COVID indicated a higher frequency of adverse reactions, we did not see any association between the history of COVID-19 and adverse reactions to the booster (Appendix A). This may be explained because all recipients of the booster received multiple antigenic stimulus (either from the previous vaccine doses or a SARS-CoV-2 infection).

There were some limitations to our study. Only 17% of HCW of the tertiary center responded to the questionnaire, in contrast to 38% in our previous study [14]. We could not assure a representative sample of the HCW population in the tertiary center. Moreover, there is the possibility that the participants with more reactogenicity answered the self-reported questionnaire. Most of our participants were female, and younger than the general population who received a booster, which is explained by the population distribution of HCW in tertiary centers. Moreover, our population was HCW, who had a higher exposure to COVID-19. There could be a possible memory bias concerning the reactogenicity of primary doses due to the time that elapsed (more than six months) between the primary series and the booster. However, the results of our questionnaire were similar to the questionnaire administered just after the primary doses. The severity of specific adverse reactions was not collected; nonetheless, we obtained data on medical leave and on possible life-threatening adverse events, which we could verify with the revision of the clinical histories of the participants.

### 4.3. Strengths and Weaknesses in Relation to Other Studies

Most of our results are consistent with those observed in previous studies that evaluated the safety of the COVID-19 booster [17,18,19]. In the COV-BOOST study, seven different COVID-19 vaccine booster doses were compared for both the BNT162b2 primary series and the ChAdOx1 nCov-19 primary series. Heterologous booster doses with mRNA-1273 did not have a higher frequency of adverse reactions compared to other heterologous and homologous booster doses [17]. Nevertheless, in a community-based prospective study, the mRNA-1273 heterologous booster doses (after the BNT162b2 or ChadOx nCoV-19 primary series) were more likely to report higher local and systemic adverse events than those receiving the BNT162b2 heterologous booster or homologous combinations [18].

In a clinical trial, when analyzing the mRNA-1273 booster in healthy adults, the safety profiles after mRNA-1273 (50 μg), mRNA-1273.351 (20 or 50 μg), and mRNA-1273.211 (50 μg) were generally similar to those observed in the primary series [20]. Other studies identified no unexpected patterns of adverse reactions after booster doses compared to those after the second dose [13,18]. However, we found a higher frequency of adenopathy in booster doses compared to the primary series. Our results are compatible with a clinical trial, where 11% (*n* = 19) of the participants reported axillary adenopathies after the administration of the mRNA-1273 booster [20].

When we reviewed the clinical histories of the participants, we did not find any serious adverse reactions after the administration of the mRNA-1273 booster. In the COV-BOOST study, six serious adverse events were deemed as possibly related to the study vaccine; none of them were related to a mRNA-1273 booster [17]. However, the majority of studies identified no serious adverse events related to the COVID-19 booster including homologous and heterologous combinations [21,22].

### 4.4. Meaning of the Study: Possible Mechanisms and Implications for Clinicians or Policymakers

Heterologous vaccination strategies have historically been applied for other vaccines, and the evidence indicates their effectiveness and safety [23]. Moreover, the evidence indicates that heterologous prime-boost vaccination has already been successfully deployed for the treatment of numerous conditions including HIV, Ebola, malaria, tuberculosis, influenza, and hepatitis B [24,25]. In a clinical trial assessing the safety and immunogenicity of the heterologous prime-boost for the Ebola virus vaccine, the standard and accelerated heterologous prime-boost regimens were well-tolerated, with no severe adverse events reported [26]. Additionally, in a clinical trial evaluating the safety and immunogenicity of a heterologous prime-boost for the influenza A/H7N7-H7N9 vaccination, the results showed a well-tolerated vaccine adverse reaction and no severe adverse events were reported following 6 months [27]. Our results are in accordance with heterologous prime-boost studies for both COVID-19 vaccines and other inmunopreventive diseases.

Few studies have compared local and systemic adverse reactions between a heterologous and a homologous booster in healthy adults. Although some clinical trials have reported a safety profile and mild reactogenicity to heterologous booster doses, there are limited data of real world studies. Our study adds to the existing data confirming that using a heterologous booster seems safe, since reactogenicity is similar to that using a homologous booster. Moreover, this is one of the first studies to explore real world data on the reactogenicity of mRNA-1273 as a booster in healthy adults.

COVID-19 booster vaccination has been authorized in Spain since October 2021. Since then, 24,071,919 booster doses have been administered up to March 2022. The Spanish pharmacovigilance system has observed 1208 adverse events, with 423 serious adverse reactions/events identified, but not related to the vaccine administration. Frequent events are mild or moderate local and systemic adverse events, and severe adverse events related to mRNA vaccines are very uncommon [28]. Our study supports the results identified in different studies and pharmacovigilance activities.

Our findings could be helpful to improve the confidence and acceptance of COVID-19 booster vaccines for vaccine recipients. Furthermore, our results may be useful to preventive medicine and occupational services to address vaccine hesitancy toward heterologous use in the general population and health care workers.

## Figures and Tables

**Figure 1 vaccines-10-01217-f001:**
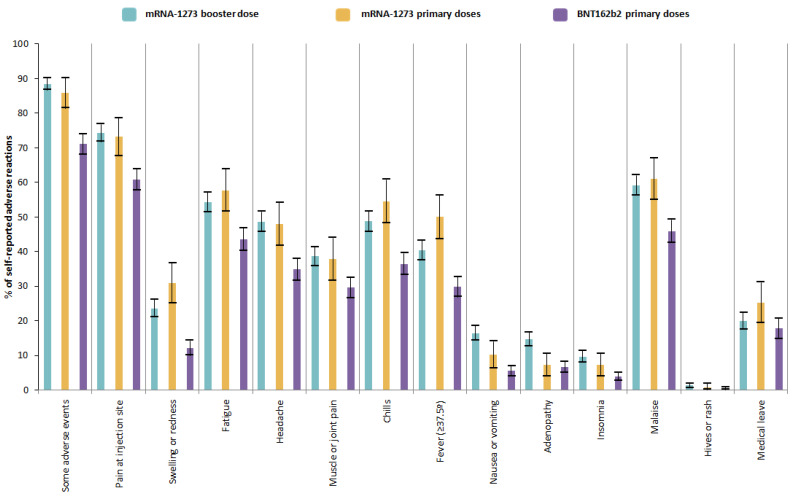
The self-reported adverse reactions to the COVID-19 vaccination when comparing the primary doses versus the booster.

**Table 1 vaccines-10-01217-t001:** The sociodemographic characteristics, medical history, and vaccination history of the sample.

Characteristics	*n* = 1222 (%)	95% CI *
Gender		
Female	1001 (81.9%)	79.8%, 84.1%
Male	221 (18.1%)	15.9%, 20.2%
Age (in years)		
Median (IQR **)	45	(33, 54)
Age group		
18–55	943 (78.1%)	75.8%, 80.5%
>55	264 (21.9%)	19.5%, 24.2%
Workers’ category		
Registered nurse	553 (45.3%)	42.5%, 48.0%
Medical doctor	284 (23.2%)	20.9%, 25.6%
Other, with patient contact	147 (12.0%)	10.2%, 13.9%
Other, without patient contact	238 (19.5%)	17.3%, 21.7%
History of allergies	54 (4.4%)	3.27%, 5.57%
History of chronic illness	135 (11.0%)	9.29%, 12.8%
History of COVID-19 infection	276 (22.6%)	20.2%, 24.9%
Seriousness of COVID-19 infection		
Asymptomatic	42 (15.2%)	11.0%, 19.5%
Mild-moderate	226 (81.9%)	77.3%, 86.4%
Hospitalization	8 (2.9%)	0.92%, 4.88%
Type of vaccine dose 1		
mRNA-1273	246 (20.1%)	17.9%, 22.4%
BNT162b2	934 (76.4%)	74.1%, 78.8%
ChAdOx1 nCoV-19	19 (1.6%)	0.86%, 2.25%
Ad26.COV2.S	5 (0.4%)	0.05%, 0.77%
Other/Unknown	18 (1.5%)	0.80%, 2.15%
Type of vaccine dose 2		
mRNA-1273	214 (18.0%)	15.8%, 20.2%
BNT162b2	961 (80.8%)	78.5%, 83.0%
ChAdOx1 nCoV-19	5 (0.4%)	0.05%, 0.79%
Other/Unknown	10 (0.8%)	0.32%, 1.36%
Type of vaccine booster		
mRNA-1273	1212 (99.2%)	98.7%, 99.7%
BNT162b2	10 (0.8%)	0.31%, 1.32%

* 95% CI: 95% confidence interval. ** IQR: interquartile range.

**Table 2 vaccines-10-01217-t002:** The self-reported adverse reactions to the COVID-19 vaccination when comparing the primary doses versus the booster.

Characteristic	mRNA-1273 Booster		mRNA-1273 Primary Doses		BNT162b2 Primary Doses		*p*-Value
	*n* = 1180 (%)	95% CI *	*n* = 246 (%)	95% CI *	*n* = 934 (%)	95% CI *	
Some adverse reaction	1044 (88.5%)	86.7%, 90.3%	211 (85.8%)	81.4%, 90.1%	664 (71.1%)	68.2%, 74.0%	<0.001
Pain at injection site	875 (74.2%)	71.7%, 76.7%	180 (73.2%)	67.6%, 78.7%	568 (60.8%)	57.7%, 63.9%	<0.001
Swelling or redness	279 (23.6%)	21.2%, 26.1%	76 (30.9%)	25.1%, 36.7%	113 (12.1%)	10.0%, 14.2%	<0.001
Fatigue	640 (54.2%)	51.4%, 57.1%	142 (57.7%)	51.6%, 63.9%	407 (43.6%)	40.4%, 46.8%	<0.001
Headache	574 (48.6%)	45.8%, 51.5%	118 (48.0%)	41.7%, 54.2%	325 (34.8%)	31.7%, 37.9%	<0.001
Muscle or joint pain	455 (38.6%)	35.8%, 41.3%	93 (37.8%)	31.7%, 43.9%	276 (29.6%)	26.6%, 32.5%	0.013
Chills	575 (48.7%)	45.9%, 51.6%	134 (54.5%)	48.2%, 60.7%	340 (36.4%)	33.3%, 39.5%	<0.001
Fever (≥37.5 °C)	476 (40.3%)	37.5%, 43.1%	123 (50.0%)	43.8%, 56.2%	278 (29.8%)	26.8%, 32.7%	<0.001
Nausea or vomiting	194 (16.4%)	14.3%, 18.6%	25 (10.2%)	6.39%, 13.9%	51 (5.5%)	4.00%, 6.92%	0.008
Adenopathy	172 (14.6%)	12.6%, 16.6%	18 (7.3%)	4.06%, 10.6%	62 (6.6%)	5.04%, 8.23%	0.070
Insomnia	115 (9.7%)	8.05%, 11.4%	18 (7.3%)	4.06%, 10.6%	37 (4.0%)	2.71%, 5.21%	0.026
Malaise	698 (59.2%)	56.3%, 62.0%	150 (61.0%)	54.9%, 67.1%	429 (45.9%)	42.7%, 49.1%	<0.001
Hives or rash	15 (1.3%)	0.63%, 1.91%	2 (0.8%)	0%, 1.94%	5 (0.5%)	0.07%, 1.00%	0.600
Medical leave	208 (19.9%)	17.5%, 22.3%	53 (25.1%)	19.3%, 31.0%	118 (17.8%)	14.9%, 20.7%	0.019
Potential life-threatening reaction	0 (0.0%)		0 (0.0%)		0 (0.0%)		

* 95% CI: 95% confidence interval.

**Table 3 vaccines-10-01217-t003:** The self-reported adverse reaction to the COVID-19 booster when comparing the homologous booster versus the heterologous booster.

Characteristic	Homologous Booster		Heterologous Booster		*p*-Value
	*n* = 246 (%)	95% CI *	*n* = 934 (%)	95% CI *	
Some adverse reaction to booster	222 (90.2%)	86.5%, 94.0%	822 (88.0%)	85.9%, 90.1%	0.300
Duration of the reaction (days, median, IQR **)	3	(2, 3)	3	(2, 3)	0.900
Pain at injection site	178 (72.4%)	66.8%, 77.9%	697 (74.6%)	71.8%, 77.4%	0.500
Swelling or redness	60 (24.4%)	19.0%, 29.8%	219 (23.4%)	20.7%, 26.2%	0.800
Fatigue	143 (58.1%)	52.0%, 64.3%	497 (53.2%)	50.0%, 56.4%	0.200
Headache	130 (52.8%)	46.6%, 59.1%	444 (47.5%)	44.3%, 50.7%	0.140
Muscle or joint pain	94 (38.2%)	32.1%, 44.3%	361 (38.7%)	35.5%, 41.8%	0.900
Chills	123 (50.0%)	43.8%, 56.2%	452 (48.4%)	45.2%, 51.6%	0.700
Fever (≥37.5 °C)	101 (41.1%)	34.9%, 47.2%	375 (40.1%)	37.0%, 43.3%	0.800
Nausea or vomiting	44 (17.9%)	13.1%, 22.7%	150 (16.1%)	13.7%, 18.4%	0.500
Adenopathy	27 (11.0%)	7.07%, 14.9%	145 (15.5%)	13.2%, 17.8%	0.072
Insomnia	18 (7.3%)	4.06%, 10.6%	97 (10.4%)	8.43%, 12.3%	0.150
Malaise	146 (59.3%)	53.2%, 65.5%	552 (59.1%)	55.9%, 62.3%	>0.999
Hives or rash	2 (0.8%)	NA, 1.94%	13 (1.4%)	0.64%, 2.14%	0.700
Medical leave after booster	57 (25.7%)	19.9%, 31.4%	151 (18.4%)	15.7%, 21.0%	0.016
Potential life-threatening reaction to booster	0 (0.0%)	0 (0.0%)	0 (0.0%)	0 (0.0%)	
Perception of booster more reactogenic	104 (46.8%)	40.3%, 53.4%	507 (61.7%)	58.4%, 65.0%	<0.001

* 95% CI: 95% confidence interval. ** IQR: interquartile range.

**Table 4 vaccines-10-01217-t004:** The odds ratio of medical leave according to the sociodemographic characteristics and the medical history variables.

Characteristics	Medical Leave		OR *	95% CI **	*p*-Value	AOR ***	95% CI **	*p*-Value
	Yes (*n* = 208)	No (*n* = 836)						
Gender								
Female	183 (88%)	682 (82%)	1.65	1.07, 2.65	0.030	1.55	0.99, 2.52	0.067
Male	25 (12%)	154 (18%)	1			1		
Age (Median and IQR **** in years)	43 (31, 52)	44 (34, 54)	0.99	0.97, 1.00	0.077	0.99	0.97, 1.00	0.034
Workers’ category								
Registered nurse	97 (47%)	375 (45%)	1			1		
Medical doctor	35 (17%)	209 (25%)	0.65	0.42, 0.98	0.043	0.77	0.49, 1.19	0.300
Other, with patient contact	34 (16%)	97 (12%)	0.74	0.47, 1.17	0.200	0.65	0.41, 1.04	0.069
Other, without patient contact	42 (20%)	155 (19%)	0.95	0.62, 1.45	0.700	0.83	0.55, 1.29	0.400
History of allergies								
Yes	17 (8%)	33 (3.9%)	2.17	1.16, 3.92	0.012	1.59	0.81, 3.01	0.200
No	191 (92%)	803 (96%)	1			1		
History of chronic illness								
Yes	34 (16%)	86 (10%)	1.70	1.10, 2.60	0.015	1.63	1.03, 2.53	0.032
No	174 (84%)	750 (90%)	1			1		
History of COVID-19 infection								
Yes	53 (25%)	188 (22%)	1.18	0.82, 1.67	0.400	1.07	0.73, 1.53	0.700
No	155 (75%)	648 (78%)	1			1		
COVID-19 vaccine booster								
Homologous booster	57 (27%)	165 (20%)	1.54	1.08, 2.17	0.016	1.45	1.00, 2.07	0.045
Heterologous booster	151 (73%)	671 (80%)	1			1		

* Odds ratio; ** 95% CI: 95% confidence interval; *** Adjusted odds ratio; **** IQR: interquartile range.

## Data Availability

The data presented in this study are available on request from the corresponding author. The data are not publicly available due to privacy restrictions.

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
