# Peer review of "Reactogenicity to the mRNA-1273 Booster According to Previous mRNA COVID-19 Vaccination"

_vaccines, 2022, doi:10.3390/vaccines10081217_

Round 1

Reviewer 1 Report

Overall the manuscript reads well. Some minor comments-

It is important to make it clear both in the Introduction section as well as in the Methods section that this is about the third dose of a COVID vaccine which was being deployed. Now that more than one COVID booster is being administered to health care workers in some regions around the world this is an important point of clarification for the reader

There do not appear to be any p values associated with Table 2, this needs to be rectified

Author Response

It is important to make it clear both in the Introduction section as well as in the Methods section that this is about the third dose of a COVID vaccine which was being deployed. Now that more than one COVID booster is being administered to health care workers in some regions around the world this is an important point of clarification for the reader

We thank the reviewer for the comments. We have included the changes proposed in the manuscript.

We have stated more clearly our definition of booster.
Material and methods, first paragraph: “For the purpose of this study, the booster was defined as a subsequent dose of vaccine administered after a COVID-19 primary series vaccination. Some participants were administered one dose of mRNA-1273 or BNT162b2 vaccine as primary series because they had a history of COVID-19 infection [11]”.

There do not appear to be any p values associated with Table 2, this needs to be rectified

We have included p values in Table 2, calculated using the Kruskal-Wallis test (non-parametric ANOVA).

Reviewer 2 Report

A comparison of the frequency of adverse reactions following the booster versus the previous vaccine doses is presented in this paper. Moreover, the comparison of the frequency of adverse reactions in those patients who received a homologous booster (previous doses of Moderna mRNA-1273 and a booster of Moderna mRNA-1273) versus those patients who received a heterologous booster (previous doses of Pfizer BioNTech BNT162b2 and a booster of Moderna mRNA-1273) is discussed. For the comparisons, authors used Pearson’s chi-squared test and Fisher’s exact test for categorical variables and Wilcoxon or Kruskal–Wallis rank sum tests for continuous variables. Data were analyzed using the statistical computing program R.

It will be good to present shortly the calculations, separately for the tests which are used, i.e. for the Pearson’s chi-squared test, the Fisher’s exact test, and Wilcoxon or Kruskal–Wallis rank sum tests. Such data will ensure that the calculations were done correctly.

Author Response

We thank the reviewer for the comment. However, we think that in our field it is uncommon to present the formula and the calculations of the tests. This is not a paper that includes new statistical methodologies; therefore, we think presenting the calculations in the paper would not be interesting for the readers. If the reviewer or the editors still think we should provide the calculations we would provide an annex file with them.  

Reviewer 3 Report

Since its emergence, SARS-CoV-2 has been a public health problem worldwide. Although several vaccines are available and many people have been vaccinated, currently circulating variants are highly transmissible. Therefore booster immunization has been recommended. The manuscript showed that the reactogenicity of mRNA vaccine as homologous and heterologous booster immunogens was not high compared to primary vaccinations. The manuscript was well prepared.

1.       Table 3: The number of “some adverse reaction” 222 is larger than total 202. Because the number of participants who got mRNA-1273 as secondary dose was 214, the number of homologous booster must not exceed 214. Please correct the table.

2.       Annex 2: Similarly, The number of “some adverse reaction” 245 is larger than total 202. Please correct the table.

3.       Based on the table 1, 276 participants experienced COVID-19 infection. Were all these infections after primary vaccination series?

Author Response

Since its emergence, SARS-CoV-2 has been a public health problem worldwide. Although several vaccines are available and many people have been vaccinated, currently circulating variants are highly transmissible. Therefore booster immunization has been recommended. The manuscript showed that the reactogenicity of mRNA vaccine as homologous and heterologous booster immunogens was not high compared to primary vaccinations. The manuscript was well prepared.

We thank the reviewer for the comments. We answer the comments point by point. We have included the changes proposed in the manuscript.

  1. Table 3: The number of “some adverse reaction” 222 is larger than total 202. Because the number of participants who got mRNA-1273 as secondary dose was 214, the number of homologous booster must not exceed 214. Please correct the table.

We have corrected the number of patients who received a booster dose with mRNA-1273, which are 264. Some participants did not receive a second COVID-19 dose because they had a history of COVID-19 infection, that’s why the total is the number of participants who received an mRNA-1273 primary dose in Table 1.

We have included an explanation in the methods section:
Material and Methods, paragraph 1: “Some participants were administered one dose of mRNA-1273 or BNT162b2 vaccine as primary series because they had a history of COVID-19 infection [11]”.

  1. Annex 2: Similarly, The number of “some adverse reaction” 245 is larger than total 202. Please correct the table.

We have corrected the totals in Annex 2.

  1. Based on the table 1, 276 participants experienced COVID-19 infection. Were all these infections after primary vaccination series?

No, we did not take into account if the COVID-19 infections happened after or before primary series vaccination. We have stated this more clearly in the article.

Material and methods, second paragraph: “The history of COVID-19 infection did not take into account if the infection happened before or after the primary series vaccination”.